# Training Language Models to Reason Efficiently

## Abstract

Scaling model size and training data has led to great advances in the performance of Large Language Models. However, the diminishing returns of this approach necessitate alternative methods to improve model capabilities, particularly in tasks requiring advanced reasoning. Large reasoning models, which leverage long chain-of-thoughts, bring unprecedented breakthroughs in problem-solving capabilities but at a substantial deployment cost associated to longer generations. Reducing inference costs is crucial for the economic feasibility, user experience, and environmental sustainability of these models.

In this work, we propose to train large reasoning models to reason efficiently. More precisely, we use reinforcement learning (RL) to train reasoning models to dynamically allocate inference-time compute based on task complexity. Our method incentivizes models to minimize unnecessary computational overhead while maintaining accuracy, thereby achieving substantial efficiency gains. It enables the derivation of a family of reasoning models with varying efficiency levels, controlled via a single hyperparameter. Experiments on two open-weight large reasoning models demonstrate significant reductions in inference cost while preserving most of the accuracy.

## 1. Introduction

Large language models (LLMs) have made significant advancements by pre-training larger models with extensive datasets (Kaplan et al., 2020), but this approach faces diminishing returns due to limited high-quality training data. An alternative to improve model capabilities, especially in domains involving careful reasoning, involves allowing models to "think" before answering, as seen in frontier reasoning models like OpenAI's o1, Gemini 2.0 Flash Thinking Experimental, and DeepSeek-R1 (Guo et al., 2025). These models produce intermediate tokens during inference, collectively referred to as *chain-of-thoughts* (Wei et al., 2022), to perform additional computations before returning an answer. The process of generating a long chain of thought before answering the user query is called *reasoning*. More precisely, *large reasoning models* with chain-of-thoughts capable of performing advanced reasoning emerge from reinforcement learning (RL) (Sutton & Barto, 2018; Guo et al., 2025) on base models using ground-truth scoring functions (e.g., correctness on math problems).

These reasoning models use test-time compute in the form of very long chain-of-thoughts, an approach that commands a high inference cost due to the quadratic cost of the attention mechanism and linear growth of the KV cache for transformer-based architectures (Vaswani, 2017). However, effective deployment of LLMs demands models that are not only accurate but also computationally efficient to serve. Even for resource-rich organizations such as large tech companies that have the resources to train reasoning models, excessive inference costs may mean operating at a loss rather than at a profit in order to match the competitor's offering. Furthermore, reducing inference costs often reduces latency, improves responsiveness, and therefore increases user experience. Finally, lowering the inference computation has a direct impact in reducing carbon emissions, with a positive benefit to both the environment and the society.

We aim to develop a procedure to *train* the model to use the appropriate amount of inference time compute to solve the problem at hand with reasoning. For straightforward problems, the resulting model would deliver efficient, direct solutions, while for more demanding tasks, it would invest additional computational effort to perform advanced reasoning. Such an adaptable model, that invests the minimum amount of compute to arrive at the correct solution, would represent a significant leap forward in terms of operational cost.

We use reinforcement learning policy gradient methods (Sutton & Barto, 2018) to train the model to use the least possible amount of tokens to reach the correct solution, thereby minimizing inference costs, ideally without compromising on accuracy. We achieve this goal by means of a modi-

---
[1]Anonymous Institution, Anonymous City, Anonymous Region, Anonymous Country. Correspondence to: Anonymous Author <anon.email@domain.com>.

Preliminary work. Under review by the International Conference on Machine Learning (ICML). Do not distribute.

fied reinforcement learning formulation which *encourages the model to produce correct answers with short chain-of-thoughts*. To the best of our knowledge, we are among the first to consider this problem, and we discuss concurrent literature in Section 2. As a result, the model learns when to stop thinking—rather than solving an easy math problem, such as simple addition, through multiple approaches, it recognizes when it has found the correct answer and concludes its reasoning efficiently while maintaining accuracy.

In order to achieve this goal, we provide a reinforcement learning implementation of the above principle which involves only a couple of line changes in a standard reinforcement learning pipeline; this allows to directly leverage existing RL codebases. Our method allows the user to *control* the reduction in inference-time compute by adjusting a scalar coefficient in an intuitive way. In other words, starting from a base reasoning model, our procedure allows to derive a *family of reasoning models*, each with increased generation efficiency (i.e., shorter chain-of-thoughts) compared to the original reasoning model.

We conduct numerical experiments on two recently released open-weight large reasoning models, DeepSeek-R1-Distill-Qwen-1.5B and DeepSeek-R1-Distill-Qwen-7B (Guo et al., 2025) and derive models with a substantial problem-dependent reduction in reasoning cost while approximately maintaining accuracy. For the 7B model, our method produces a model with a reduction of 16% tokens on the competition-level benchmark American Invitational Mathematics Examination 2024 while slightly increasing accuracy, and a reduction of 30% with a small reduction in accuracy of 1% on the MATH dataset (Hendrycks et al., 2021), and a reduction of approximately 50% tokens on GSM8K (Cobbe et al., 2021b) with similar accuracy, thereby showing the ability of the model to dynamically reduce its test-time compute budget with minimal loss in accuracy.

Beyond its simplicity, an attractive property of our approach is its *computational efficiency*: although training reasoning models with large scale reinforcement learning may have a prohibitive cost (Guo et al., 2025), our procedure shows that training them to reason efficiently is highly viable even with modest academic resources: our models are obtained with only 100 reinforcement learning steps (approximately 200 gradient updates). The fact that we achieve a performance comparable to that of the original reasoning model with a short training is surprising, because in few RL steps the model needs to optimize for reasoning patterns that are shorter and more efficient than the original model.

## 2. Related Work

**Improving model capabilities with test-time compute** Several techniques have been developed to enhance LLM

reasoning through more test-time compute. **Chain of thoughts** (Wei et al., 2022) can be seen as one such fundamental method. **Prompt engineering** is a broadly applicable technique (White et al., 2023) which can be used to elicit specific abilities that are thought to be useful to reach a solution, such as thinking step by step, exploring multiple solution paths, and double-checking the answer. However, it does not scale because it does not train the model to use these strategies effectively. **Self consistency** (Wang et al., 2022) on the other hand is one of the most effective ways to enhance test-time performance when test-time verifiers are not available. The method generates multiple final answers and then returns the mode of their empirical distribution. As the mode of the empirical distribution converges to the mode of the population level distribution of the model answers, the method does not scale well with the number of samples, and moreover, it is only effective when the answers can be clustered together, such as in math problems. This limitation can be bypassed by **Best-of-N**, a simple, general purpose and effective search technique. It relies on sampling multiple responses from the model and then selecting the best at test time according to the scoring function; however, it critically relies on the availability of an accurate test-time scoring function (Gao et al., 2023). A more sophisticated search technique is **Monte Carlo Tree Search**, because it directs the compute budget to the most promising directions of the search space. It was a critical component to the development of AlphaGo (Silver et al., 2017). However the algorithm is not directly applicable outside of structured search frameworks. **Tree-of-thoughts** (Yao et al., 2024) and its extension (Gandhi et al., 2024; Besta et al., 2024) can be seen as implementing search in natural language but they are limited by their bespoke nature. **Process reward models** (Lightman et al., 2024) provide step by step numerical guidance on the progress of the chain-of-thought, but they have not been as effective to build large scale reasoning systems. Finally, **self-correction** (Kumar et al., 2024) trains the LLMs to fact-check itself; however, it implements a specific technique within a scripted framework rather than being a general purpose technique to enhance the reasoning capabilities with more test-time compute.

While the above mentioned techniques can be highly effective in specialized scenarios, modern large scale reasoning models, which we discuss next, are trained with reinforcement learning and rely on autoregressive generation.

**Large Reasoning Models** Frontier reasoning such as OpenAI o1, Deepseek R1 and QwQ-preview rely on long, monolithic chain-of-thoughts to perform advanced and general purpose reasoning. They are trained with large scale reinforcement learning (Guo et al., 2025), which leads them to develop emerging abilities, such as branching, verification and backtracking. Our approach aims at making these

models more efficient to deploy.

**Efficient serving**   While we focus on developing reasoning models that can be served efficiently, our approach is orthogonal to existing methods from the literature of efficient LLMs; see Zhou et al. (2024) for a recent survey. For example, system-level techniques build a system to accelerate inference. Some examples include speculative decoding (Leviathan et al., 2023) and batch engines like vLLM (Kwon et al., 2023a); both can be directly combined with our method. Model-based techniques, on the other hand, act directly on the model to accelerate inference. Some examples include weight pruning (Liu et al., 2018) and quantization (Lin et al., 2024), which can also be combined with our methodology. In contrast, our approach leverages reinforcement learning to train the model for computational efficiency, making it applicable whenever the chain of thought is not required in the final answer.

**Concurrent works**   To our knowledge, the first open-weight LLM that can be classified as a 'reasoning' model—producing long monolithic chain of thoughts—is the 32 billion parameter model QwQ-preview, which was released on November 28 on the Hugging Face. As these models are very recent, we are not aware of prior studies on efficiently training these models to reason efficiently except for some concurrent work, which we review below.

Chen et al. (2024) investigate the overthinking phenomena and propose methods to mitigate it by using heuristics such as First-Correct Solutions (FCS) and Greedy Diverse Solutions (GDS) to generate preference data which is then used for offline policy optimization. However, this method doesn't allow easily tuning the model to the user's compute budget. The concurrent technical report of Kimi k1.5 (Team et al., 2025) also reports a method to shorten the chain-of-thought using a length penalty in the reward function while doing online RL, a procedure similar in principle but not identical to ours. We note that their procedure does not appear to have a tunable parameter which allows to obtain a family of models–each with varying trade-offs–as we do.

## 3. Setup

Let $p$ be a language model. When provided with a prompt $x$, the language model produces a response $y = (y^1, y^2, ..., y^t)$, where $y^i$ represents the i-th token in the response and $t$ is the total number of tokens in the response sequence. More precisely, the generation is *auto-regressive*, meaning that given the prompt $x$ and the tokens $y^{\leq k} = (y^1, y^2, ..., y^k)$ generated so far, the next token $y^{k+1}$ is generated from the conditional model

$$y^{k+1} \sim p(\cdot \mid x, y^{\leq k}). \tag{1}$$

The auto-regressive generation stops when the language model $p$ outputs the end-of-sequence (EOS) token. Therefore, if $y = (y^1, y^2, ..., y^t)$ is a full response, $y^t$ is always the EOS token. With a little abuse of notation, we also let $y \sim p(\cdot \mid x)$ denote the process of sampling the full response $y = (y^1, y^2, ..., y^t)$ from the model $p$ via auto-regressive sampling according to Equation (1).

**Chain-of-Thoughts**   Chain of thoughts, introduced by (Wei et al., 2022), is a key framework to implement reasoning. Given a prompt $x$, the LLM is said to produce a "chain of thoughts" when it produces intermediate tokens that are not part of the output before generating the final answer in an autoregressive way. Typically, the final answer is not formally separated from the chain-of-thoughts, and so we let $z$ denote the full output of the model $z \sim p(x)$.

**Objective function and reinforcement learning**   We consider problems where the responses generated from an LLM can be evaluated by a scoring function $f(x, z) \mapsto \mathbb{R}$, often called *reward model* or *verifier*, that measures the suitability of the response. For math problems, such as those that we consider in this paper, the reward function establishes whether the solution to the problem is correct (Cobbe et al., 2021b; Hendrycks et al., 2021)

$$f(x, z) = 1\{z = z^\star(x)\} \tag{2}$$

where $z^\star(x)$ is the correct answer to the math problem $x$. Since $z$ is the full output of the model, including the chain of thought, the relation $z = z^\star(x)$ tests whether the final answer generated by the model coincides with the gold answer, rather than checking the equivalence between strings.

Large reasoning models (Guo et al., 2025) are reportedly trained with reinforcement learning (Sutton & Barto, 2018). When a chain of thoughts is used, the objective function to maximize can be written as

$$\text{ACCURACY}(p) = \mathbb{E}_{x \sim \rho} \mathbb{E}_{z \sim p(x)} \big[ 1\{z = z^\star\} \big]. \tag{3}$$

where $\rho$ is the prompt distribution. In the sequel, we simply write $\mathbb{E}$ to denote the expectation. For math problems, maximizing Equation (3) directly maximizes the probability that the model correctly solves a random question the prompt distribution.

## 4. Method

We aim to design a method that trains models to use the minimum amount of inference time compute to arrive at the correct answer. For simpler math problems, such as those in GSM8K (Cobbe et al., 2021b), the model should recognize when it has reached the correct solution within a few hundred tokens. In contrast, for competition-level problems like

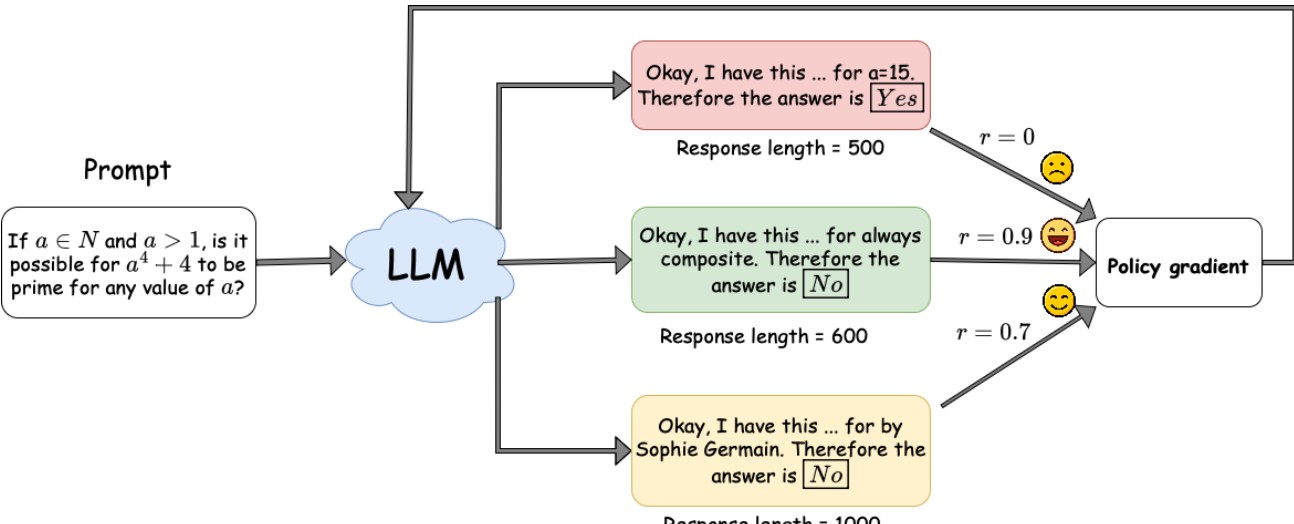

Figure 1. Pipeline depicting our method. For every prompt, multiple solutions are sampled and rewarded based on correctness and response length. The shortest correct answers are rewarded the highest and the language model is then updated using policy gradients.

AIME, the model should be capable of expending thousands of tokens if that is necessary to find a strategy that solves these exceptionally challenging questions.

One attractive option is to train the model on an objective function derived from Equation (3) that *encourages the model to produce correct solutions with the minimum amount of tokens*. In order to achieve the latter goal, we penalize the length of the correct responses

$$\mathbb{E}\Big[1\{z = z^\star(x)\}(1 - \alpha f(\text{LEN}(z)))\Big] \qquad (4)$$

using a monotonic function $f$ of the input and a tunable parameter $\alpha \in [0, 1)$. The choice $\alpha = 0$ yields the reinforcement learning objective (3); increasing $\alpha$ increases the regularization towards shorter—but correct—responses.

In order to ensure that the length regularization is effective, we first normalize the length of the responses and then use the sigmoid function $\sigma$ to soft-clip it, obtaining

$$f(\text{LEN}(x)) = \sigma\left(\frac{\text{LEN}(z) - \text{MEAN}(x)}{\text{STD}(x)}\right) \qquad (5)$$

where

$$\text{MEAN}(x) = \mathbb{E}_{\substack{y \sim p(x), \\ \text{s.t. } 1\{z=z^\star\}=1}} \big[\text{LEN}(z)\big]$$

$$\text{STD}(x) = \sqrt{\text{Var}_{\substack{z \sim p(x), \\ \text{s.t. } 1\{z=z^\star\}=1}} \big[\text{LEN}(z)\big]}$$

are the *per-prompt* mean and standard deviation of the length, respectively. The per-prompt normalization ensures

that longer chains of thought on hard problems are not disproportionately penalized compared to shorter ones on easier problems. When $\alpha \in [0, 1)$, the sigmoid ensures that the objective function is always bounded between $[0, 1]$ even for abnormally long generations, and that correct responses, even if long, are always preferred to incorrect ones.

### 4.1. Optimizing the objective with Reinforcement Learning

Since optimizing Equation (4) involves sampling from the model auto-regressively, the objective function is non-differentiable; however, it can be optimized with reinforcement learning, for instance with policy gradient methods (Sutton & Barto, 2018).

One popular option is proximal policy optimization (PPO) (Schulman et al., 2017) which considers the (local) objective function

$$\min\{f_\theta^t(y, x)\mathcal{A}(y^{<t}, x), \text{clip}_{1-\epsilon}^{1+\epsilon}(f_\theta^t(y, x)\mathcal{A}(y^{<t}, x)\}$$

defined using the density ratio

$$f_\theta^t(y, x) = \frac{\pi_\theta(y^t | x + y^{<t})}{\pi_{old}(y^t | x + y^{<t})}$$

and for a suitable choice for the advantage estimator $\mathcal{A}(y^{<t}, x)$. Traditionally, in deep reinforcement learning (Schulman et al., 2017) the advantage estimator involves a neural network.

With language models, maintaining a separate value network to obtain a variance-reduced advantage estimator (Schulman et al., 2017) may add significant computational and

implementation complexity without necessarily increasing performance (Kool et al., 2019; Ahmadian et al., 2024). One simple and effective alternative is to just estimate the advantage using Monte Carlo (MC) as proposed by (Kool et al., 2019; Ahmadian et al., 2024). Such estimator is also called REINFORCE Leave One Out (RLOO) estimator. To be precise, the trajectory advantage can be estimated as

$$\mathcal{A}(y_i, x) = \mathcal{R}(y_i, x) - \frac{1}{n-1} \sum_{j \neq i} \mathcal{R}(y_j, x)$$

where $\mathcal{R}$ is the trajectory return and $y_i$ is the $i$ generation for prompt $x$. We then simply use the sequence level advantage as the token level advantage, namely $\mathcal{A}(y^{<t}, x) = \mathcal{A}(y, x)$. In essence, we use PPO with the RLOO advantage estimator.

# 5. Experiments

We seek to evaluate our method through numerical experiments. In particular, we aim to answer the following questions:

- What are the trade-offs between accuracy and inference cost?

- What are simple baselines that are relevant in this setting?

We first discuss the setup, then introduce some baselines, present the empirical results and associated trade-offs, and finally discuss some ablations.

## 5.1. Setup

**Initial unsuccessful experiments**  In our initial experiments, we performed distillation from QwQ-32B-Preview to Qwen2.5-3B-Instruct and Qwen2.5-1.5B-Instruct so as to elicit strong reasoning skills in these two models; these distilled models would have served as a starting point for our method.

However, to our surprise, the distilled models showed a regression in performance on common benchmarks such as MATH and AIME 2024 compared to the instruct model, despite using much longer chain-of-thoughts with qualitatively more advanced reasoning patterns. Although our method is still effective in reducing the length of the chain-of-thought of the distilled model, these experiments do not accurately reflect the trade-off between inference-cost and accuracy when the instruct model is also taken into account.

**Models and Datasets**  We revisited our method following the release of the reasoning models DeepSeek-R1-Distill-Qwen-1.5B and DeepSeek-R1-Distill-Qwen-7B (Guo et al., 2025). These models were distilled from the more powerful DeepSeek-R1 using industry-grade techniques. Along with

a LLaMA-variant distilled by the same authors (Guo et al., 2025), they are the only open-weight reasoning models of their size. Notably, they demonstrate impressive performance on challenging benchmarks such as AIME 2024.

For post-training the model using our technique, we choose 3.2k prompts from the MATH, cn_k12, AIME, AoPS and the Olympiad subsets of the Numina Math dataset (LI et al., 2024). The dataset includes problems that lack an objective answer, such as proof-based questions. We filter out such problems and ensure that the selected training problems have a numerical answer that can be parsed. We use the same dataset across all baselines to ensure consistency.

**Evaluation**  We evaluate the models on three datasets, ordered by increasing difficulty:

- GSM8K (Cobbe et al., 2021a), which contains grade-school-level math problems,

- MATH (Hendrycks et al., 2021) which is a standard benchmark containing harder problems than GSM8K,

- the The American Invitational Mathematics Examination (AIME) 2024, a competition-level set of challenging mathematical problems.

For all models, we set the temperature to 0.6 as suggested in the model's card[1] and set the token limit to 32K. We use vLLM (Kwon et al., 2023b) for efficient batch inference. We use the parser created by the Qwen Team for the evaluation of their models[2] to measure correctness.

We report the *average pass rate@k* for all models. Specifically, for each prompt, we sample $k$ responses and compute the average accuracy per prompt, which is then averaged across the entire dataset. For GSM8K, we set $k = 1$ due to its large number of test samples. In contrast, for MATH500, we use $k = 3$, and for AIME2024, we set $k = 10$ given its limited set of only 30 questions.

**Implementation details**  We build on the OpenRLHF codebase (Hu et al., 2024). For the 1.5B model, we use 4 GH200 GPUs on one low-density node and for the 7B model, we use 8 GH200 GPUs distributed across two low-density nodes (4 GPUs per node). We set vLLM to the maximum context length (32K) during generation and set the generation temperature to 1. For training the 1.5B, ZeRO Stage 2 (Rajbhandari et al., 2020) is used and for the 7B, ZeRO Stage 3 with activation checkpointing is required to prevent out of memory errors. The training precision is set to bfloat16. We generate 8 responses for each prompt. For

---

[1] https://huggingface.co/deepseek-ai/DeepSeek-R1-Distill-Qwen-7B

[2] https://github.com/QwenLM/Qwen2.5-Math

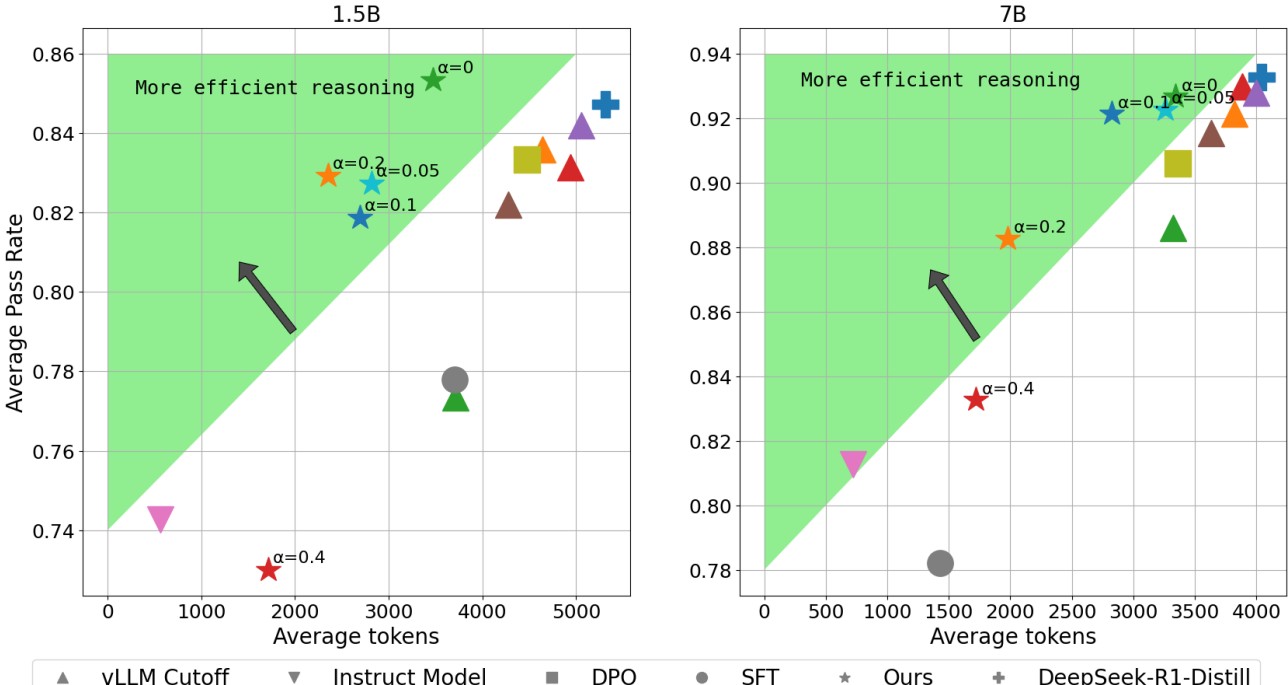

Figure 2. This figures describes the results of our training on the MATH500 test set. The green triangle in the left corner represents the desirable trend of models where higher accuracy is achieved with a lower number of tokens.

every iteration, 32 prompts are selected from the dataset and the global batch size is set to 128 which leads to 2 gradient steps per RL iteration. For the 1.5B, the learning rate is set to $5 \cdot 10^{-6}$ and for the 7B, it is set to $2 \cdot 10^{-6}$. For all experiments, Adam (Kingma & Ba, 2017) is used as the standard optimizer. We experiment with 4 values of $\alpha$: $0.05, 0.1, 0.2$ and $0.4$. For all RL experiments, the value of the KL coefficient is set to $10^{-3}$. We use the same prompt template for all models which can be found in Appendix C.

### 5.2. Baselines

To the best of our knowledge, apart from the concurrent work discussed in Section 2, no prior studies have explored this setting. Alongside our method, we introduce and implement simple baseline approaches that help balance inference cost and accuracy.

1. **Generation Cutoff:** This simple baseline imposes a maximum token limit during the vLLM generation. If a response exceeds the token limit and remains incomplete, it is assigned a score of 0. We evaluate token cutoffs at 8,000, 16,000, 20,000, 24,000, and 32,000.

2. **Rejection Sampling + SFT:** In this baseline, we generate 8 solutions per prompt using the distilled 1.5B and 7B models. From the generated solutions, we

select the shortest correct responses. For a dataset of 3,200 prompts, this process yields approximately 2,200 and 2,500 valid responses for the 1.5B and 7B models, respectively. We experiment with three learning rates: $1 \times 10^{-5}$, $5 \times 10^{-6}$, and $2 \times 10^{-6}$. We find that $5 \times 10^{-6}$ effectively reduces response length in a meaningful way.

3. **DPO:** Using the same dataset as above, we select response pairs consisting of the longest and shortest correct solutions and apply Direct Preference Optimization (DPO) (Rafailov et al., 2023) on these preference pairs. While other preference optimization algorithms are applicable in this setting, we choose DPO for its popularity and ease of use. Similar to the SFT baseline, we experiment with three learning rates: $1 \times 10^{-5}$, $5 \times 10^{-6}$, and $2 \times 10^{-6}$. We observe that $1 \times 10^{-5}$ effectively reduces response length, whereas the other rates do not achieve any reduction.

### 5.3. Numerical Results

We train DeepSeek-R1-Distill-Qwen-1.5B and DeepSeek-R1-Distill-Qwen-7B models using different values of $\alpha \in [0, 0.05, 0.1, 0.2, 0.4]$ to illustrate the trade-offs between models with different lengths for the chain-of-thoughts, and report the results on MATH500 in Figures 2 and 3. The re-

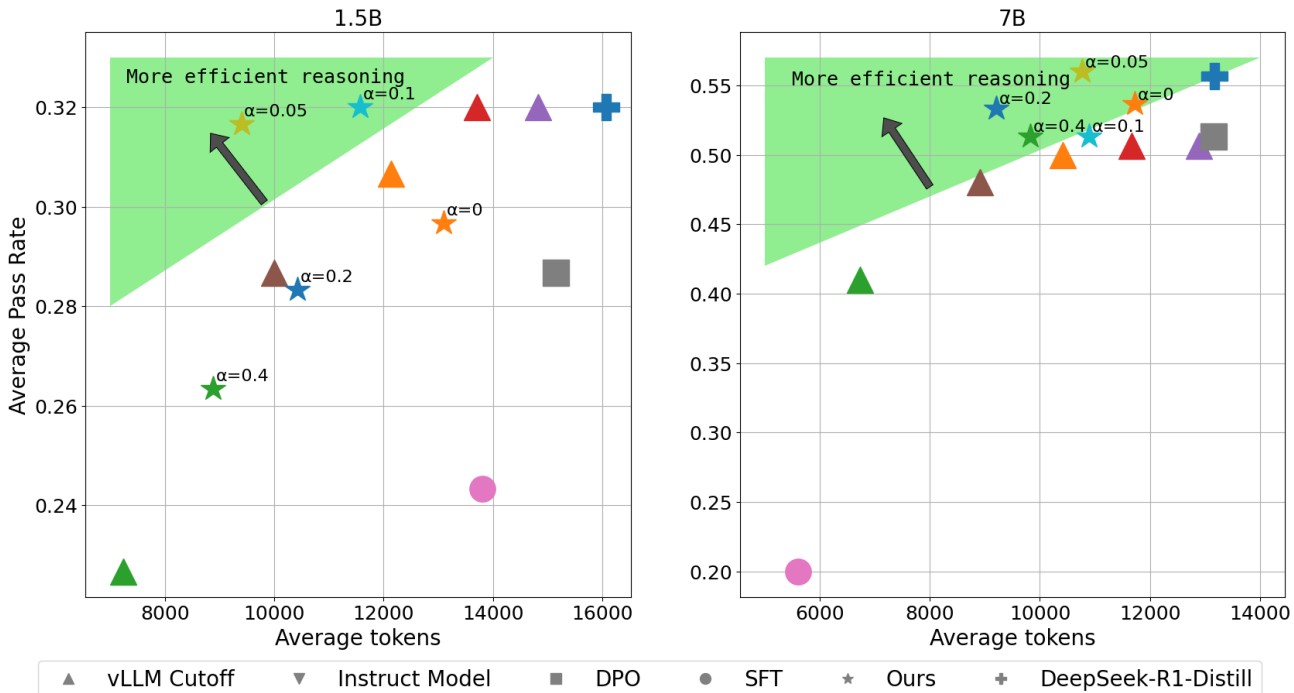

*Figure 3.* Results on AIME2024 dataset

sults for GSM8K can be found in Appendix B due to space reason.

As shown in Figure 2, our method enables smooth trade-offs of compute cost and accuracy, allowing models to be tailored to the specific requirements of downstream tasks or users based on different values of $\alpha$. For instance, with $\alpha = 0.1$, the length of the chain-of-thought of the 7B model on the MATH dataset decreases by 30% (from 4000 to 2800 tokens) while the accuracy loss is only 1%. Similarly, in the AIME dataset (Figure 3), setting $\alpha = 0.2$ reduces token usage by 30% (from 14,000 to 9,000) while incurring only a 2% accuracy drop compared to the DeepSeek-R1-Distill-Qwen-7B.

We offer several remarks:

- We prompt the original DeepSeek-R1-Distill-Qwen-7B and one of our models about a simple question "How much is 1+1?". While DeepSeek-R1-Distill-Qwen-7B reasoning model expends several tokens (more than a page in the appendix) to arrive at the correct solution, the model trained with our method quickly reaches the same conclusion within few tokens.

- The models trained with our procedures adapt the length of the chain of thought to the difficulty of the problem. For example, $\alpha = 0.2$ brings a token saving of 22% on AIME24 and of 77% on GSM8K compared

to doing RL at $\alpha = 0$.

- Even without any length penalty (i.e., $\alpha = 0$), we observe a reduction in response length on both the MATH and AIME datasets. We hypothesize that this occurs because these models have not been previously trained with reinforcement learning (RL) and have only undergone a single round of distillation from R1.

- The SFT and DPO baselines appear to perform worse than early-stopping the vLLM generation.

- The model with the highest $\alpha = 0.4$ experiences a larger performance drop compared to the others. In Figure 4, we visualize the training dynamics by plotting accuracy every 10 RL iterations. The figure illustrates how $\alpha = 0.4$ induces a rapid reduction in response length, likely preventing the model from adapting effectively, ultimately leading to lower performance.

### 5.4. Ablations

We perform an ablation to study a highly critical design component in the implementation of our method, namely the decision of not normalizing the advantage function in the RL training procedure.

In fact, it is a standard practice (e.g., (Shao et al., 2024)) to

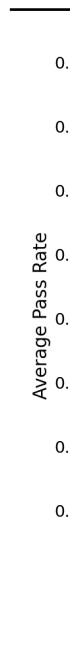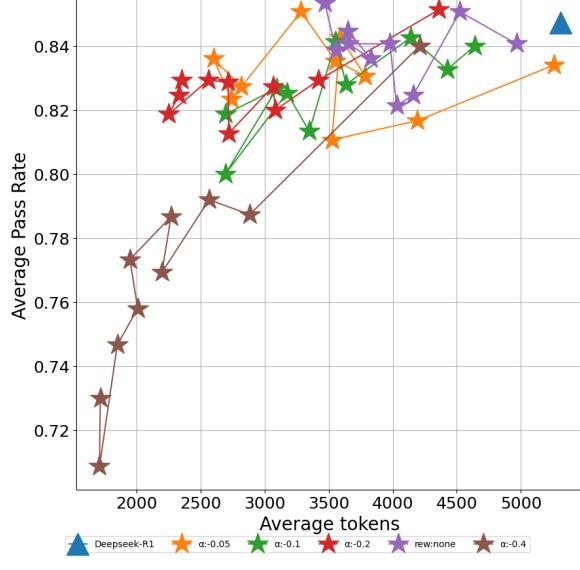

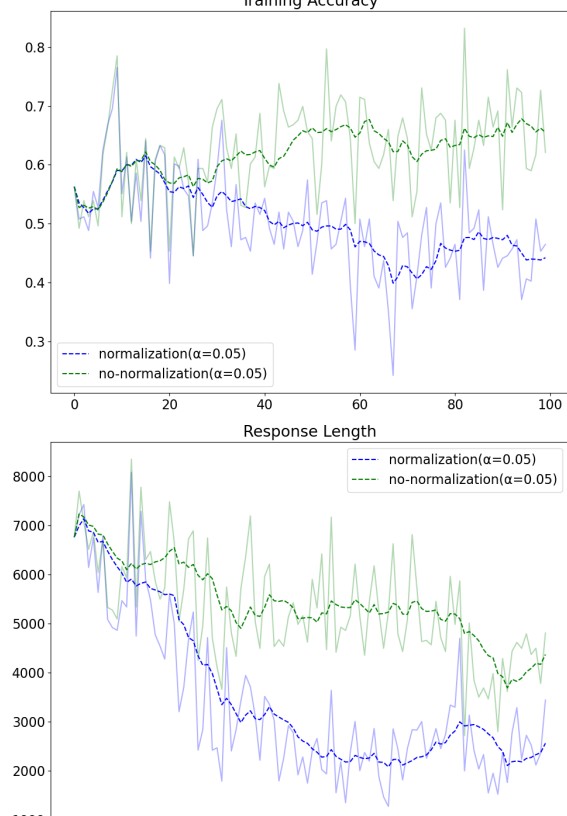

*Figure 4.* Evolution of the model during training with varying values of $\alpha$. Checkpoints are created after 10 iterations.

normalize the token-level advantage function and obtain

$$\hat{A}_{i,t} = \frac{r_i - r_{mean}}{r_{std}}$$

where $r_{mean}$ is the mean reward and $r_{std}$ is the standard deviation of the rewards. While this choice is sensible in a more standard setting, it can have unintended consequences when the objective function contains the length penalty.

Consider the case where for a prompt $x$, all responses are correct. In that case, all rewards will be distributed within $[1 - \alpha, 1]$. Assume that the reward distribution is uniformly distributed in $[1 - \alpha, 1]$. In that case, the mean reward is $1 - \frac{\alpha}{2}$ and the standard deviation is $\frac{\alpha}{\sqrt{12}}$. The normalized advantage value for a correct response with maximum value $r = 1$ (i.e., the shortest correct response) becomes $\frac{1 - (1 - \alpha/2)}{\frac{\alpha}{\sqrt{12}}} = \sqrt{3}$ which is independent of $\alpha$! In other words, the advantage normalization, under certain conditions, can bring a length decrease independent of $\alpha$. The resulting length decrease is generally too substantial for the model to absorb, and this leads to a sharp drop in accuracy during training, as can be seen in Figure 5. .

## 6. Limitations

Our optimization procedure, while effective, is somewhat more involved than SFT or DPO-derived techniques because of the reinforcement learning setup. Furthermore, the choice of the penalty coefficient $\alpha$ affects the overall generation cost but does not precisely target a precise generation length, which may be required by some latency-constrained

*Figure 5.* Advantage normalization rapidly decreases the response length alongside accuracy.

applications. We leave such exact controllability as future work.

## 7. Conclusion

In this work, we introduced a novel methodology that significantly reduces the inference cost for reasoning models while minimally affecting its accuracy. Our approach is related in spirit to model distillation; however, rather than reducing deployment cost by reducing the model size, we focus on reducing the deployment cost by reducing the inference cost of the same model.

A key advantage of our framework is its ability to adapt computational resources based on problem difficulty. This suggests that rather than training separate models targeting various inference-time compute trade-offs, a single model can adjust its inference budget dynamically. This property holds promise for applications requiring scalable, cost-effective AI solutions that are highly efficient without compromising on accuracy.

## 8. Impact Statement

This paper presents work whose goal is to advance the field of Machine Learning. There are many potential societal consequences of our work, none which we feel must be specifically highlighted here.

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

## A. Qualitative examples of the model before and after training

We prompt the Distilled 7B model and our model trained with $\alpha = 0.4$ on a simple prompt.

Prompt: Please reason step by step, and put your final answer within \boxed{}. Question: What is 1+1?

**Model (before training):**

<think> Okay, so I need to figure out what 1 plus 1 is. Hmm, let me think about this. I remember in school, they taught us that adding numbers is like combining groups of things. So, if I have one apple and someone gives me another apple, how many apples do I have? I guess that's two apples. Wait, let me try to visualize it. Imagine I have one block here and another block there. If I put them together, how many blocks do I have in total? That's right, two blocks. So, 1 plus 1 equals 2. But hold on, is this always the case? Like, in different number systems or something? I mean, I know in binary, 1 plus 1 is 10, but I think that's a different topic. The question is about regular addition, right? So, in standard arithmetic, 1 plus 1 is definitely 2. I can also think about the number line. If I start at 1 and move one step forward, where do I land? That's 2. So, that makes sense too. Maybe I can use my fingers to check. If I hold up one finger and then hold up another finger, how many fingers am I showing? Two fingers. Yep, that confirms it. Is there a more mathematical way to look at this? I remember something about addition being the simplest arithmetic operation. It's the basis for more complex calculations. So, 1 plus 1 is the first step in understanding addition. I also recall that in set theory, the sum of two sets with one element each results in a set with two elements. So, if set A has one element and set B has one element, then the union of A and B has two elements. That's another way to see that 1 plus 1 equals 2. Wait, but in some programming languages, like Python, what happens if I write 1 +

1? Let me think. If I open a Python interpreter, and type 1 + 1, it should output 2. Yeah, that's correct. So, even in code, it's 2. Is there any context where 1 plus 1 doesn't equal 2? Like in different mathematical structures or something? Hmm, in Boolean algebra, sometimes 1 plus 1 is 1, but that's not standard arithmetic anymore. It's more like a logical OR operation. But the original question is about regular addition, so that's not applicable here. Also, in some contexts like in a group where the operation is defined differently, 1 + 1 might not be 2, but again, that's beyond the scope of basic arithmetic. I guess another way to think about it is using the Peano axioms, which are the foundation of arithmetic. According to these axioms, 1 is a natural number, and the successor of 1 is 2. So, adding 1 to 1 gives you the successor of 1, which is 2. That's pretty solid. Maybe I can also use real-life examples. If I have a glass with one liter of water and I pour another liter into it, how much water do I have? Two liters. So, that's another practical demonstration that 1 plus 1 equals 2. I wonder if there's a geometric interpretation. If I have a line segment of length 1 and another line segment of length 1, and I place them end to end, the total length would be 2. Yep, that works too. So, after thinking through all these different angles|counting, number lines, sets, programming, Boolean algebra, axioms, real-life examples, and geometry|it all consistently points to the fact that 1 plus 1 equals 2. There doesn't seem to be any situation within the realm of standard arithmetic where this isn't true. I guess another way to reinforce this is by using objects I have at hand. Like, if I have one book and someone gives me another book, how many books do I have? Two books. Or if I have one toy car and find another toy car, now I have two toy cars. This practical application really helps solidify the concept. I also remember learning

605 about the commutative property of
606 addition, which states that the order
607 of numbers being added doesn't change
608 the result. So, 1 + 1 is the same
609 as 1 + 1, which is still 2. And the
610 associative property, which allows us
611 to add more than two numbers together
612 without changing the result. Although,
613 in this case, we're only adding two
614 numbers, so that might not apply
615 directly, but it's good to know for
616 more complex problems. Additionally,
617 thinking about money could help. If I
618 have one dollar and I receive another
619 dollar, how much money do I have? Two
620 dollars. That's another real-world
621 example supporting the idea that 1 plus
622 1 equals 2. I suppose another way to
623 view it is through tally marks. If
624 I make one tally mark and then make
625 another tally mark, how many marks do
626 I have? Two tally marks. So, that's
627 another visual method to confirm it. I
628 also recall that in some early number
629 systems, like Roman numerals, 1 is
630 represented by 'I', and adding another
631 'I' gives you 'II', which stands for
632 two. So, even in different numeral
633 systems, the concept holds. Thinking
634 about the history of mathematics,
635 the concept of 1 plus 1 equals 2 was
636 established long ago. Euclid defined
637 numbers and their operations in his
638 "Elements," laying the foundation
639 for arithmetic as we know it today.
640 So, this basic principle has been
641 consistent throughout history. In
642 conclusion, after considering all these
643 various perspectives|counting, number
644 lines, sets, programming, Boolean
645 algebra, axioms, real-life examples,
646 geometry, and even the history of
647 math|it's clear that 1 plus 1 equals
648 2 in standard arithmetic. There's
649 no ambiguity or alternative result
650 in these different contexts. So, I
651 can confidently say that the answer is
652 2. </think> 1 plus 1 equals 2. This
653 is consistently true across various
654 mathematical contexts, from basic
655 arithmetic to more complex structures.
656 $\boxed{2}$
657
658
659

**Model (after training, $\alpha = 0.4$):** <think> To solve (1 + 1), I start by identifying the two numbers involved, which are both 1. Next, I add these two numbers together. Finally, the sum of 1 and 1 is 2. </think> To solve 1 + 1, follow these simple steps: 1. **Identify the numbers to add:** Both numbers are 1. 2. **Perform the addition:** 1 + 1 = 2 Therefore, the final answer is $\boxed{2}$.

## B. Results on GSM8K

The results on the GSM8K test set are in Figure 6. Its interesting that the DeepSeek model performs worse than the Instruct model on the GSM8K test set. However, with our RL training, we are able to get it to the same performance. However, it is hard to make it more compute efficient than the Instruct model in this dataset.

## C. Prompt template for training

For all training purposes, we use the following prompt template:

Please reason step by step, and put your final answer within \boxed{}. Question: *QUESTION*

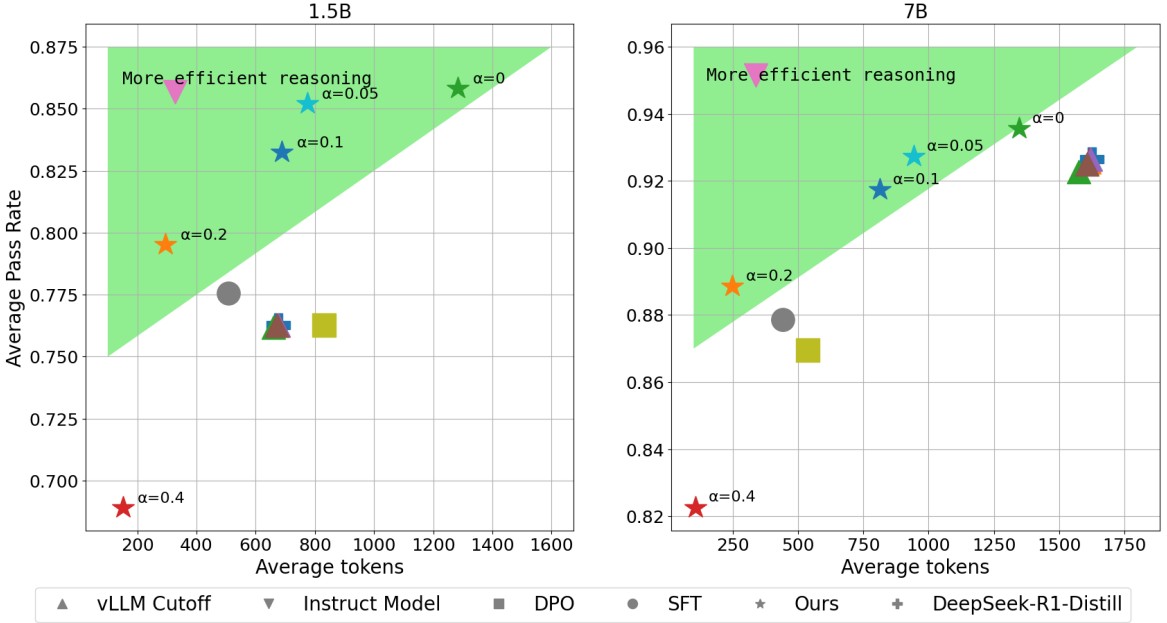

*Figure 6.* This figures describes the results of our training on the GSM8k test set. The green triangle in the left corner represents the desirable trend of models.

