# OpenReview forum: "Training Large Language Models to Reason Efficiently"
_ICML.cc/2025/Conference — Submitted to ICML 2025_

### Official Review · Reviewer_nXze · 2025-03-10

**Overall Recommendation:** 2

**Summary:**

The paper proposes an approach to finetune reasoning models to reduce unnecessary reasoning steps while preserving accuracy. The approach penalizes excessive reasoning steps while ensuring the model still arrives at correct answers. Experimental results show that the proposed RL achieves up to 50% token reduction while not sacrificing too much accuracy on some benchmarks.

**Claims And Evidence:**

The claim about computational efficiency is not convincing to me. Since the proposed method needs to fine-tune the model using RL, which is already a high computational cost step. Besides, the choice of $\alpha$ value seems to require extensive experiments, thus leading to a significant computational cost.

**Essential References Not Discussed:**

N/A

**Experimental Designs Or Analyses:**

The choice of baselines can be better from my perspective. It would be more insightful to compare the proposed RL approach against current LRMs. For instance, how would response length and accuracy differ if we applied a DeepSeek Zero-style RL approach versus the proposed RL objective?

**Methods And Evaluation Criteria:**

Yes.

**Other Comments Or Suggestions:**

N/A

**Other Strengths And Weaknesses:**

S1. The paper is easy to understand.

S2. Reducing the reasoning tokens is an interesting direction.

W1. My major concern is about the choice of $\alpha$. The proposed method is very simple. It just adds one more regularization term of response length to the RL objective. The results show that the proposed method is very sensitive to $\alpha$ values. In this case, how to choose the value of $\alpha$ is very important. Include a more descent way to select $\alpha$ would be much better.

W2. The penalty coefficient allows reducing inference cost globally but does not enforce exact token limits.

W3. The proposed method is only evaluated on mathematical reasoning benchmarks (GSM8K, MATH, AIME). It would be better to see the method's effectiveness on other domains, such as logical inference.

**Questions For Authors:**

See Other Strengths And Weaknesses

**Relation To Broader Scientific Literature:**

The paper shows that RL has good potential to increase the reasoning ability of LLMs with fewer tokens.

**Theoretical Claims:**

There is no theoretical claim.

---

> ### Author Rebuttal · Authors · 2025-03-31
>
> We thank the reviewer for their thoughtful and constructive feedback. We would like to address their main concerns and clarify a few points raised in the review.
>
> We acknowledge the concern about computational cost. However, we believe it is important to distinguish between training and deployment. The cost of training is amortized over a large number of inference calls, as training is a one-time process. Our method aims to significantly reduce inference-time compute, which can be the dominant cost at scale in real-world deployments.
>
> **Comparing against current LRM:**
>
> We appreciate the reviewer’s suggestion regarding comparison with methods such as DeepSeek-Zero. However, we believe there may be a misunderstanding here. DeepSeek-Zero focuses on training new reasoning models from base LLMs using verifiable outcomes, whereas our method is designed to post-train existing reasoning models. Our goal is not to improve reasoning capability from scratch, but to make already capable models more efficient at inference time while preserving accuracy. Thus, the objectives and use cases of the two approaches are fundamentally different.
>
> **W1: Choice of $\alpha$**
>
> We thank the reviewer for highlighting this important point. The sensitivity to the $\alpha$ parameter is indeed a feature, not a flaw. Our method is deliberately designed to offer flexibility—rather than outputting a single model, it yields a family of models with different efficiency-accuracy trade-offs which can be obtained by varying $\alpha$. This allows users to select a model that best fits their application needs, whether they prioritize cost savings or accuracy.
>
> **W2: Not enforcing exact token limits**
>
> We appreciate the reviewer’s suggestion of enforcing strict token limits. We considered this design choice, but found that such constraints can lead to brittle behavior. Real-world problems vary in complexity, and harder problems naturally require more reasoning steps. Our approach encourages adaptive computation: the model spends fewer tokens on easier problems while allocating more tokens to harder ones, all while maintaining high accuracy. This adaptive behavior is a key strength of our method.
>
> **W3: Other benchmarks**
>
> Thank you for this suggestion. Following the reviewer's recommendation, we evaluated our method on CommonSenseQA [1] and the Logical Deduction task from BIG-Bench [2], which are out-of-distribution compared to our original math benchmarks. The results are available at: https://imgur.com/a/FEbdRL7. The plots demonstrate that our method generalizes to prompts that are out of distribution such as problems in CommonSenseQA and Logical Deduction. For instance, for $\alpha=0.2$, we get 40% reduction in tokens but only 1.1% drop in accuracy for the 7B model on CommonSenseQA. Similarly, for Logical Deduction with $\alpha=0.2$, we get a 50.7% reduction in number of tokens but only 3.5% drop in accuracy on Logical Deduction.
>
>
> **References**
>
> [1] CommonsenseQA: A Question Answering Challenge Targeting Commonsense Knowledge by Talmor et al. [https://aclanthology.org/N19-1421/]
>
> [2] Beyond the Imitation Game: Quantifying and extrapolating the capabilities of language models by Srivastava et al. [https://arxiv.org/abs/2206.04615]

---

### Official Review · Reviewer_kuvk · 2025-03-13

**Overall Recommendation:** 4

**Summary:**

This paper presents a simple but effective way to reduce the reasoning length of o1/r1 like RL-based reasoning models without any inductive bias. The method is rather clear and simple but effective.

**Claims And Evidence:**

Yes.

**Essential References Not Discussed:**

No.

**Experimental Designs Or Analyses:**

I buy most part of this paper, which is a generalizable way to achieve significant reasoning length reduction.

**Methods And Evaluation Criteria:**

Classic RL based reasoning evaluations: GSM8K, MATH and AIME, with rl scaling curves on response length and accuracy. Appropriate settings.

**Other Comments Or Suggestions:**

No

**Other Strengths And Weaknesses:**

No explicit weaknesses. Though more fruitful analysis including more model families and sizes would be appreciative, they are not necessary. The important part is that it actually works to reduce the reasoning length in a general way (by only changing the reward mixture) without compromising performance.

**Questions For Authors:**

No

**Relation To Broader Scientific Literature:**

RL for LLM reasoning.

**Theoretical Claims:**

no

---

> ### Author Rebuttal · Authors · 2025-03-31
>
> We thank the reviewer for the positive and encouraging feedback. We're glad the clarity, simplicity, and effectiveness of our method came through, and appreciate your recognition of its generalizability.
>
> Should the reviewer have any further questions, we would be happy to discuss them.

---

### Official Review · Reviewer_tgK5 · 2025-03-14

**Overall Recommendation:** 3

**Summary:**

The paper proposes a reinforcement learning approach that trains models to dynamically allocate inference-time computation based on task difficulty. By incorporating a length penalty into the reward function—with a tunable hyperparameter α—the method encourages the model to produce correct answers with shorter reasoning chains when possible. Experiments on math problem datasets (including GSM8K, MATH, and AIME2024) demonstrate that the approach can substantially reduce the number of generated tokens with minimal impact on accuracy. The paper also compares several baselines and provides ablation studies on key design choices.

**Claims And Evidence:**

The paper claims that by introducing a length penalty into the RL objective, it is possible to train reasoning models that maintain accuracy while significantly reducing the computational cost during inference, and the experiment result including token usage and pass rate comparisons on datasets such as GSM8K, MATH500 and AIME2024—support this claim.

**Essential References Not Discussed:**

To my knowledge, there are no essential references not discussed.

**Experimental Designs Or Analyses:**

The study is conducted on three math problem datasets (GSM8K, MATH, AIME2024) and includes comparisons with several baselines such as generation cutoff, rejection sampling combined with SFT, and DPO. The design clearly shows how different values of α affect token count and accuracy.

 The experiments are well-designed to illustrate the trade-offs. However, the evaluation is mostly confined to math problems. Extending the analysis to other reasoning or natural language tasks would help assess the method’s generalizability

**Methods And Evaluation Criteria:**

The proposed methods and evaluation criteria are appropriate.  And the authors choosed the up-to-date open-source reasoning LLMs such as QwQ-Preview and R1-Distilled sereis to conduct experiments,  and they choose challenging benchmark such as AIME2024 which serves as harder problems and GSM8K which serves as simpler problems. It well aligns with the research topic and purpose.

**Other Comments Or Suggestions:**

1. Consider expanding the experimental evaluation to include non-mathematical reasoning tasks to demonstrate broader applicability.

2. Provide a more detailed sensitivity analysis of the hyperparameter α across different datasets and tasks.

**Other Strengths And Weaknesses:**

Strengths:

1. The paper addresses a practical issue—high inference cost in reasoning models—with a novel, easy-to-integrate solution.

2. The method leverages a single hyperparameter (α) to control the efficiency-accuracy trade-off, which is a clear and intuitive design.

3. The experimental results are thorough and clearly illustrate the benefits of the approach.

Weaknesses:

1. The experiments are primarily limited to math reasoning tasks, which raises questions about the method’s applicability to other domains.

2. The inherent instability and sensitivity of RL training may make replication challenging; a deeper discussion on this aspect would be beneficial.

3. The theoretical analysis of the reward function and length penalty, while insightful, remains somewhat preliminary and could be expanded.

**Questions For Authors:**

Questions:

1. How might your method generalize to other reasoning domains beyond mathematical reasoning? Would you expect similar efficiency gains for tasks requiring commonsense reasoning or logical reasoning?

2. Your results show that even with α = 0 (no explicit length penalty), you observe a reduction in response length on MATH and AIME datasets. You hypothesize this occurs because the models haven't been previously trained with RL. Could you elaborate on this hypothesis and discuss whether multiple rounds of RL might yield further improvements?

**Relation To Broader Scientific Literature:**

The key contributions of the paper are related to current research of reasoning LLMs, which includes o1/o3 like LLMs along with DeepSeek-R1 series and Qwen-QwQ, the proposed methods could help current version of LRMs to better reduce the computatioanl cost and strike a balance between reasoning cost and the reasoning accuracy.

**Theoretical Claims:**

The paper presents a mathematical formulation of their modified RL objective and explains how it incentivizes shorter reasoning chains while preserving accuracy. The formulation appears sound, particularly the normalization approach to ensure balanced penalties across problems of varying difficulty.

---

> ### Author Rebuttal · Authors · 2025-03-31
>
> We sincerely thank the reviewer for their thoughtful feedback. We appreciate that the reviewer recognized the practical relevance of our work and the simplicity of our proposed solution.
>
> **[1] Evaluating on non-math tasks.**
>
> We appreciate the suggestion to evaluate the method on tasks beyond mathematical reasoning. Following this, we conducted experiments on CommonSenseQA [1] (commonsense reasoning) and Logical Deduction from BIG-Bench [2] (logical reasoning). Results are available at https://imgur.com/a/FEbdRL7. These plots indicate that our approach generalizes well to out-of-distribution prompts.
> For example, on CommonSenseQA, with $\alpha = 0.2$, we observed a 40% reduction in tokens with only a 1.1% drop in relative accuracy using the 7B model. On Logical Deduction, the same $\alpha$ value led to a 50.7% token reduction and just a 3.5% drop in accuracy. These results support the broader applicability of our method beyond the math domain.
>
>  **[2] On RL instability**
>
> Thank you for highlighting this important concern. We fully agree that reproducibility and stability are critical in RL-based methods. We are committed to open-sourcing all our code and training configurations to facilitate replication. In our experience, training has been stable across multiple runs. Training dynamics are visualized at [https://imgur.com/a/SxN5Id3](https://imgur.com/a/SxN5Id3), and we did not observe any unexpected divergence or instability.
>
>  **[3] Theoretical discussion around the reward**
>
> We acknowledge that the theoretical underpinnings of length-penalized reward functions are still developing. Our current formulation represents an initial attempt to explore this trade-off space. One open question we raise for future work is whether a Pareto-optimal reward function exists that more effectively balances accuracy and efficiency. We hope this paper serves as a stepping stone for deeper theoretical exploration in this area.
>
>  **[4] Why length decreases when using $\alpha=0$**
>
> This was indeed an intriguing observation for us as well. Recent work by Liu et al. [3] points to a bias in the GRPO loss function: it averages per-token loss across entire sequences, which unintentionally favors shorter correct sequences over longer correct ones, and longer incorrect sequences over shorter incorrect ones. This may explain the unexpected reduction in reasoning length, even when $\alpha=0$.
> We tested the fix proposed in [3] and observed that the length reduction disappears when applying it. Table 1 below shows normalized accuracy and token usage (relative to a baseline 7B distilled model). The average results highlight that the fix mitigates the unintended length bias.
>
> Table 1: Table showing the effects of fixes proposed in [3]. $\Delta$ NT refers to change in normalized tokens. $\Delta$ NA refers to change in normalized accuracy. All numbers are normalized based on the Baseline scores. All experiments have been conducted on the 7B Distilled model.
>
> Dataset | RLOO+Fix($\alpha=0$) ($\Delta$NT) | RLOO + Fix ($\alpha=0$) ($\Delta$NA) | RLOO($\alpha=0$) ($\Delta$NT) | RLOO($\alpha=0$) ($\Delta$NA) | Baseline (NT) | Baseline (NA) |
> |-----------|-----------------|----------------|-----------|----------|---------------|---------------|
> |MATH500 |  2.3               |     -0.4           |     -17.4      |    -0.6      |      100         |     100          |
> |AIME2024 |     8            |         -3       |       -10.9    |    -3.6      |     100          |    100           |
> |GSM8k  |   -12.2              |      -3.37          |    -17.2       |   1.08       |    100           |    100           |
> |Average| **-0.64**      |   -2.25 | -15.16 | -1.04 | 100 | 100 |
>
>
>
>
>
> **References**
>
> [1] CommonsenseQA: A Question Answering Challenge Targeting Commonsense Knowledge by Talmor et al. [https://aclanthology.org/N19-1421/]
>
> [2] Beyond the Imitation Game: Quantifying and extrapolating the capabilities of language models by Srivastava et al. [https://arxiv.org/abs/2206.04615]
>
> [3] Understanding R1-Zero-Like Training: A Critical Perspective, Liu et al. [https://arxiv.org/pdf/2503.20783]

---

### Official Review · Reviewer_5weZ · 2025-03-17

**Overall Recommendation:** 2

**Summary:**

* The paper proposes a training procedure to find reasonable trade-offs of accuracy-compute to solve a reasoning problem.
  * "accuracy" in terms of mathematical reasoning abilities (e.g., GSM8K benchmark)
  * "compute" in terms of average inference-time tokens with CoT required to answer the question
* The crux of the training procedure is a reward model that penalizes the length of the response. Apart from this, the training appears fairly standard using PPO with Leave one out estimator.
* The approach is evaluated on standard mathematical reasoning benchmarks - GSM8K, MATH, AIME. Results suggest that in some cases, one can observe reasonable accuracy-compute trade-offs e.g., on MATH, decreasing average token rate by 30% for a 1% drop in accuracy.

**Claims And Evidence:**

Generally, claims and evidence is somewhat convincing.

**Essential References Not Discussed:**

* The paper overlooks 2x very important directions, many of which I believe should be baselines
* **Direction 1: Efficient/Compressed CoT**
  * Nayab, Sania, et al. "Concise thoughts: Impact of output length on llm reasoning and cost." arXiv preprint arXiv:2407.19825 (2024).
  * Han, Tingxu, et al. "Token-budget-aware llm reasoning." arXiv preprint arXiv:2412.18547 (2024).
  * Xia, Heming, et al. "Tokenskip: Controllable chain-of-thought compression in llms." arXiv preprint arXiv:2502.12067 (2025).
    * This one is specifically after the ICML submission deadline. I do not factor this for the rating.
* **Direction 2: Compute optimal test-time strategy** many of which are from Q4 2024
  * Snell, Charlie, et al. "Scaling llm test-time compute optimally can be more effective than scaling model parameters." ICLR '25 (arXiv August '24)
  * Bansal, Hritik, et al. "Smaller, weaker, yet better: Training llm reasoners via compute-optimal sampling." ICLR '25 (arXiv August '24)

**Experimental Designs Or Analyses:**

Yes, I checked the soundness/validity of experimental designs. This is quite standard e.g., using pass rates in typical mathematical reasoning benchmarks.

**Methods And Evaluation Criteria:**

Method makes sense for the problem i.e., adding a penalty to discourage number of tokens used to arrive at a solution. Evaluation criteria (i.e., pass rate, average tokens) is reasonable too.

**Other Comments Or Suggestions:**

Some nitpicks:
- Fig. 2, 3:
   - Generally difficult to read.
   - Unclear what the colors of the symbols mean.
   - Clarify what criteria is used to shade the green region. Caption says "desirable" -- but what exactly is desirable?
- L273 "... distilled ... using industry-grade techniques": what techniques specifically?

**Other Strengths And Weaknesses:**

### Strengths

1. The paper is well-motivated: reasoning incurs a drastic increase in inference-time compute due to number of tokens used.
2. Some results are promising e.g., on MATH, decreasing average token rate by 30% for a 1% drop in accuracy.

### Concerns

**1. (Major) Missing crucial baselines / discussions of prior works**
* (extends remarks in "Essential References Not Discussed")
* I believe some important baselines are missing for comparison, which have previously been shown to be competitive
  * Prompted truncation
  * "TALE" Token-Budget-Aware LLM Reasoning Code [appears to be available](https://github.com/GeniusHTX/TALE)

**2. (Major) Missing discussion on why some experiments were unsuccessful**
* The paper discusses on experiments with Qwen2.5-{1.5, 3}B models and observed a regression in performance. As a result, the paper moves to experiments on Deepseek-R1 models.
* Without any additional discussion, this observations suggests that the proposed method works on some models, and not on others, for unknown reasons.
* I highly recommend the authors to address this discrepancy. Because otherwise, the results appear cherry-picked to cater to a subset of models where the approach worked.

**3. "Dynamic" allocation of inference-time compute**
* There are multiple claims that refer to dynamic allocation of compute (e.g., L27, L89, L435).
* In light of multiple prior works (e.g., [Snel et al., ICLR '25]) that refer to *dynamic* allocate compute depending on prompt and compute budget, I would argue that the proposed approach is not dynamic -- given that one cannot allocate a token budget at test-time.
* I recommend authors to either remove the "dynamic" claims, or carefully define "dynamic" in relation to prior works.

**Questions For Authors:**

Please see comments under "Strengths And Weaknesses" -- especially the ones listed as major concerns.

**Relation To Broader Scientific Literature:**

* The key contribution of the paper is enabling efficient inference for mathematical reasoning tasks.
* Existing approaches largely train models to use CoT reasoning chains, without restricting the size of these chains.

**Theoretical Claims:**

No theoretical claims in the paper.

---

> ### Author Rebuttal · Authors · 2025-03-31
>
> We thank the reviewer for the thorough and insightful comments, which have greatly helped us improve the clarity and rigor of our manuscript.
>
> Below, we address the reviewer’s major concerns:
>
> [1] **Missing baseline**:
>
> We appreciate the reviewer highlighting the missing baseline. However, both the 'prompted truncation' and 'TALE' baselines **rely on the assumption that the language model can effectively follow explicit instructions**, such as "Respond in less than 500 tokens." Our empirical findings suggest that smaller reasoning-focused models (e.g., the DeepSeek distilled models used in our experiments) lack robust instruction-following capabilities. Consequently, as demonstrated in the tables below, there is no meaningful correlation between the instructed token limit and the actual length of generated responses:
>
> Table 1: Number of tokens generated for varying token limits using Distilled-R1-Qwen-1.5B on MATH500:
>
> | Token Limit | Tokens Generated |
> |-------------|------------------|
> | 256         | 4609.34          |
> | 512         | 4915.71          |
> | 768         | 5228.85          |
> | 1024        | 4913.84          |
> | 1280        | 5306.68          |
> | 2048        | 5064.06          |
> | 4096        | 5245.11          |
>
> Table 2: Number of tokens generated for varying token limits using Distilled-R1-Qwen-7B on MATH500:
>
> | Token Limit | Tokens Generated |
> |-------------|------------------|
> | 256         | 3434.56          |
> | 512         | 3587.05          |
> | 768         | 3518.34          |
> | 1024        | 3716.17          |
> | 1280        | 3524.46          |
> | 2048        | 3688.01          |
> | 4096        | 3815.11          |
>
> Prompt used:
> > "Please think step by step and answer in less than X tokens. Question: {question} Answer:"
>
> Given this limitation, our proposed method offers an advantage by not relying on explicit token-length instructions, ensuring broader applicability and effectiveness for models that lack reliable and general instruction-following abilities.
> We will add the comparison with these baselines in the next iteration of the manuscript.
>
> We appreciate the reviewer’s suggestion regarding the additional baselines and will include them in our citations. However, we believe that a direct comparison with some of these works may fall outside the scope of this paper. For example, Snell et al. examine scenarios involving parallel sampling from the LLM, whereas Bansal et al. focus on the training aspect.
>
> [2] **Clarification regarding unsuccessful experiments**:
>
> We apologize for the confusion caused by our previous phrasing. To clarify, our initial exploratory experiments focused on fine-tuning smaller instruct models (Qwen2.5-1.5B and Qwen2.5-3B) using extended reasoning demonstrations from QwQ-32B-Preview. However, these fine-tuned models unexpectedly showed decreased performance compared to their instruct counterparts:
>
> | Model                           | MATH500 Performance |
> |---------------------------------|---------------------|
> | Qwen2.5-1.5B-Instruct           | 55.2                |
> | Qwen2.5-1.5B-Instruct + SFT     | 44.7                |
> | Qwen2.5-3B-Instruct             | 65.9                |
> | Qwen2.5-3B-Instruct + SFT       | 61.3                |
>
> This finding aligns with previously reported observations in the literature [1], suggesting that such fine-tuning may negatively impact smaller instruct-model performance. Due to this challenge, we postponed further experiments until the recent release of highly performant small-scale reasoning models by DeepSeek [2]. The superior capabilities of these new models provided a suitable foundation to test and validate our proposed method effectively.
>
> [3] **Use of the term "dynamic"**:
>
> We apologize for any ambiguity caused by our use of the adjective "dynamic." Originally, our intention was to highlight the adaptive nature of the response-length reduction. To ensure clarity, we will omit the word "dynamic" and instead explicitly state that reductions in response length are more pronounced for easier problems and less so for harder ones.
>
> [4] **Minor suggestions**:
>
> We thank the reviewer for pointing out these minor but important details. We will carefully implement these corrections and improvements, significantly enhancing the manuscript's clarity and readability.
>
> We appreciate the reviewer’s valuable feedback, which has notably strengthened our manuscript.
>
>
> **References**
>
> [1] LIMR: Less is More for RL Scaling by Li et al. [https://arxiv.org/pdf/2502.11886]
>
> [2] DeepSeek-R1: Incentivizing Reasoning Capability in LLMs via Reinforcement Learning
> [https://arxiv.org/abs/2501.12948]

---

### Decision · Program_Chairs · 2025-05-01

**Decision:**

Reject

**Comment:**

This paper studies a simple approach to improve the reasoning ability of models. In particular, the paper introduces a reward which encourages the model to generate correct solutions with fewer tokens. The paper also provides empirical results to support the claim. The reviews for the paper are borderline with primary concerns of the reviewers being: (1) missing baseline comparisons and (2) sensitivity to hyperparameter. Thank you to the authors for their rebuttal responding to the reviewers' concerns. However, Reviewer 5weZ  & nXze have raised questions about sensitivity of the approach to the hyperparameter and missing baseline comparisons, to which the authors' response was not convincing to the reviewers.

After going through the paper, I think these concerns are somewhat valid. Firstly, I think there is a need to add appropriate baseline comparison to the paper (even in cases where the authors think the baseline could possibly be weak). Second, while authors dismissed Reviewer nXze comment on sensitivity on alpha as a feature (mentioning it provides tradeoff between accuracy & response length), it is not completely evident if alpha provides such a clear tradeoff from Figure 2 & 3 of the paper. Overall, I think this paper is on the borderline.